# *Arid1b* haploinsufficient mice reveal neuropsychiatric phenotypes and reversible causes of growth impairment

Cemre Celen[1,2†], Jen-Chieh Chuang[1,2†], Xin Luo[1,2,3], Nadine Nijem[4,5], Angela K Walker[6,7], Fei Chen[1,4,6], Shuyuan Zhang[1,2], Andrew S Chung[1,2], Liem H Nguyen[1,2], Ibrahim Nassour[1,2], Albert Budhipramono[1,2], Xuxu Sun[1,2], Levinus A Bok[8], Meriel McEntagart[9], Evelien F Gevers[10], Shari G Birnbaum[7], Amelia J Eisch[11], Craig M Powell[6,7], Woo-Ping Ge[1,4,6], Gijs WE Santen[12], Maria Chahrour[4,5], Hao Zhu[1,2*]

[1]Children's Research Institute, University of Texas Southwestern Medical Center, Dallas, United States; [2]Departments of Pediatrics and Internal Medicine, Center for Regenerative Science and Medicine, University of Texas Southwestern Medical Center, Dallas, United States; [3]Department of Bioinformatics, University of Texas Southwestern Medical Center, Dallas, United States; [4]Departments of Neuroscience, University of Texas Southwestern Medical Center, Dallas, United States; [5]Eugene McDermott Center for Human Growth and Development, University of Texas Southwestern Medical Center, Dallas, United States; [6]Departments of Neurology and Neurotherapeutics, University of Texas Southwestern Medical Center, Dallas, United States; [7]Department of Psychiatry, University of Texas Southwestern Medical Center, Dallas, United States; [8]Department of Pediatrics, Máxima Medical Center, Veldhoven, The Netherlands; [9]Medical Genetics, St George's University Hospitals, NHS Foundation Trust, United Kingdom Caroline Brain, Endocrinology, Great Ormond Street Hospital for Children, London, United Kingdom; [10]William Harvey Research Institute, Barts and the London, Queen Mary University of London, London, United Kingdom; [11]Department of Anesthesiology and Critical Care Medicine, Children's Hospital of Philadelphia, and Mahoney Institute of Neuroscience, Perelman School of Medicine, University of Pennsylvania, Philadelphia, United States; [12]Department of Clinical genetics, Leiden University Medical Center, Leiden, The Netherlands

**\*For correspondence:** Hao.Zhu@ utsouthwestern.edu

[†]These authors contributed equally to this work

**Competing interests:** The authors declare that no competing interests exist.

**Abstract** Sequencing studies have implicated haploinsufficiency of *ARID1B*, a SWI/SNF chromatin-remodeling subunit, in short stature (Yu et al., 2015), autism spectrum disorder (O'Roak et al., 2012), intellectual disability (Deciphering Developmental Disorders Study, 2015), and corpus callosum agenesis (Halgren et al., 2012). In addition, *ARID1B* is the most common cause of Coffin-Siris syndrome, a developmental delay syndrome characterized by some of the above abnormalities (Santen et al., 2012; Tsurusaki et al., 2012; Wieczorek et al., 2013). We generated *Arid1b* heterozygous mice, which showed social behavior impairment, altered vocalization, anxiety-like behavior, neuroanatomical abnormalities, and growth impairment. In the brain, *Arid1b* haploinsufficiency resulted in changes in the expression of SWI/SNF-regulated genes implicated in neuropsychiatric disorders. A focus on reversible mechanisms identified Insulin-like growth factor (IGF1) deficiency with inadequate compensation by Growth hormone-releasing hormone (GHRH) and Growth hormone (GH), underappreciated findings in *ARID1B* patients. Therapeutically, GH supplementation was able to correct growth retardation and muscle weakness. This model

functionally validates the involvement of *ARID1B* in human disorders, and allows mechanistic dissection of neurodevelopmental diseases linked to chromatin-remodeling.

## Introduction

It is becoming clear that SWI/SNF chromatin-remodeling complexes have a major impact on human diseases, from cancer to neuropsychiatric disorders to body size regulation (*Kadoch and Crabtree, 2015*; *Ronan et al., 2013*). SWI/SNF chromatin-remodeling complexes use the energy of ATP to remodel nucleosome density and position to control epigenetic states, lineage differentiation, and cellular growth during development and cancer (*Kadoch and Crabtree, 2015*; *Ho and Crabtree, 2010*). ARID1B is a 236 kDa protein that contains an AT-rich DNA interactive domain ('ARID' domain) and facilitates proper genomic targeting of ARID1B containing SWI/SNF complexes. *ARID1B* is the most commonly mutated gene in Coffin-Siris syndrome (CSS), a monogenic syndrome characterized by growth retardation, facial dysmorphism, and intellectual disability (ID) (*Santen et al., 2012*; *Tsurusaki et al., 2012*; *Wieczorek et al., 2013*). In addition, *ARID1B* is among the most frequently mutated genes in autism spectrum disorders (ASD) and non-syndromic ID (*O'Roak et al., 2012*; *Deciphering Developmental Disorders Study, 2015*). In these diseases, *ARID1B* mutations are scattered across the gene without clear accumulation in particular domains (*Ronan et al., 2013*; *Santen et al., 2013*). Since these are often predicted to be nonsense or frame-shift mutations, heterozygous *ARID1B* loss-of-function is hypothesized to be the causative genetic mechanism. Up to this point, if and how *ARID1B* mutations translate into various human phenotypes is unknown, and animal models have not yet been used to model or devise novel treatments for these '*ARID1B*-opathies'.

Here, we employ genetically engineered mouse models to elucidate the phenotypic impact of *Arid1b* mutations. We developed an *Arid1b* haploinsufficient mouse that exhibits neuropsychiatric abnormalities reminiscent of ASD, as well as the developmental and growth retardation phenotypes seen in CSS. Although not previously considered a cardinal feature of this syndrome, a meta-analysis of 60 patients by Santen *et al.* shows that on average, stature is considerably shortened in CSS patients by about two standard deviations (*Santen et al., 2013*). After showing the clinical relevance of our mouse model, we focused on potentially reversible etiologies of behavioral and growth phenotypes. We observed GHRH-GH-IGF1 axis deficiencies in *Arid1b* heterozygous mice and also found evidence for this in humans. GH supplementation in mice rescued growth retardation and muscle weakness, which are also salient features of human *ARID1B*-opathies. Though successful in *Mecp2* mutant mice that model Rett syndrome (*Tropea et al., 2009*; *Castro et al., 2014*), intervening on the GH-IGF1 axis was not able to reverse neuropsychiatric defects associated with *Arid1b*. Our findings not only functionally validate *ARID1B's* involvement in human disease, they suggest underappreciated clinical manifestations of human *ARID1B* mutations that can be approached from a treatment-perspective.

## Results

### Characterization of *Arid1b⁺ᐟ⁻* and *Arid1b⁻ᐟ⁻* mice

Using Cas9 germline gene-editing, we generated whole-body knockout and conditional floxed mice (*Figure 1A*). *Arid1b* is prominently expressed in the cortex, cerebellum, and hippocampus (*Lein et al., 2007*; *Ka et al., 2016*) (*Figure 1—figure supplement 1A*). In heterozygous mice, *Arid1b* mRNA transcripts were reduced in liver, whole brain, pituitary gland, dentate gyrus, and hypothalamus (*Figure 1B*). Protein levels showed a similar pattern in whole brain extracts, and homozygous P0 pups showed an absence of ARID1B protein (*Figure 1C*). Homozygous mice were born but died perinatally (*Figure 1D*). To model the genetics of haploinsufficient human ARID1B-opathies, we generated whole-body heterozygous (*Arid1b⁺ᐟ⁻*) mice [birth ratios from *Arid1b⁺ᐟ⁻* x *Arid1b⁺ᐟ⁺* crosses: 389/661 (58.9%) WT, 272/661 (41.1%) *Arid1b⁺ᐟ⁻*], which survived into adulthood and appeared healthy but were small for age (*Figure 1E*). There were no abnormalities in electrolytes, liver function tests, or blood counts (*Figure 1—figure supplement 1B–D*). 16/272 (6.6%) of *Arid1b⁺ᐟ⁻* mice had hydrocephalus, the displacement of brain parenchyma by accumulated cerebrospinal fluid, a

**eLife digest** DNA does not just float freely inside our cells. Instead, it is wound around proteins called histones and packaged tidily into a form called chromatin. This packaging allows genes to be switched on or off by making it easier or harder to access different stretches of the genetic code.

A group of proteins called the SWI/SNF chromatin-remodeling complex are responsible for the packing and unpacking of DNA during development, dictating the fate of thousands of genes. Mutations that affect one component of this complex, a protein known ARID1B, are associated with a rare genetic condition called Coffin-Siris syndrome, and may also have a role to play in autism spectrum disorders and intellectual disability. However, there were previously no animal models that can be used to study this mutation in the laboratory.

Celen, Chuang et al. have now genetically modified mice to remove one of their two copies of the gene that encodes the mouse equivalent of ARID1B. This change replicates the mutation that is most commonly seen in people with Coffin-Siris syndrome. Celen, Chuang et al. report that the mutant mice with just one working copy of the gene showed many features also seen in Coffin-Siris syndrome, including a smaller size and weaker muscles. The mutant mice also repeated certain behaviors, like grooming themselves, and showed unusual interactions with other mice.

Further tests showed that the mutant mice had lower than expected levels of growth hormone in their blood. The mice were then treated with growth hormone supplements to find out if this could reverse any of their symptoms. Indeed, this treatment made the mice larger and stronger, but did not change their behavior.

Some doctors are already treating people with Coffin-Siris syndrome with growth hormone, and these new findings suggest that this treatment counteracts defects caused directly by the mutation affecting ARID1B. Moreover, this mouse model will allow the role of ARID1B to be investigated further in the laboratory, and could be used as a tool to discover, develop and test new treatments for Coffin-Siris syndrome.

condition that frequently accompanies Dandy-Walker malformations seen in CSS patients (*Schrier Vergano et al., 2013*) (*Figure 1—figure supplement 1E*).

## *Arid1b*$^{+/-}$ mice developed abnormal social, vocal, and behavioral phenotypes

Given the associations between *Arid1b* mutations and ASD, we examined behaviors related to this disorder. To examine social interactions, we quantified the time spent interacting with a juvenile target mouse. Compared to WT littermate controls, *Arid1b*$^{+/-}$ mice spent significantly less time interacting with unfamiliar juvenile mice (*Figure 1F*), suggesting impaired social behavior. To enrich the connections between *Arid1b*$^{+/-}$ mice and ASD-like phenotypes, we also performed grooming and marble burying tests that examined repetitive behaviors (*Silverman et al., 2010*). Consistent with other ASD mouse models, *Arid1b*$^{+/-}$ mice exhibited increased self-grooming (*Figure 1G*) and potentially as a consequence, buried less marbles (*Figure 1—figure supplement 2A*). A similar pattern of repetitive behaviors was seen with *Synapsin* knockout mice, another mouse model of ASD (*Greco et al., 2013*).

Another feature of ASD is abnormal communication and language. Several mouse models of ASD and language disorders show alterations in one or more vocalization parameters, including the number, duration, frequency, amplitude, and other characteristics of ultrasonic vocalizations (USVs) (*Konopka and Roberts, 2016*; *Araujo et al., 2015*). Furthermore, ASD patients who have retained speech tend to exhibit abnormalities in voice quality and pitch (*Kanner, 1968*; *Bonneh et al., 2011*). USVs emitted by *Arid1b*$^{+/-}$ mice are longer in duration, and have abnormal pitch (*Figure 1H,I* and *Figure 1—figure supplement 2B*). Interestingly, *Arid1b*$^{+/-}$ mice emitted the same total number of USVs compared to WT mice (*Figure 1—figure supplement 2C*), suggesting altered modulation rather than absent vocalizations.

Anxiety-like behavior, a comorbidity of ASD, was examined using three separate tasks in male mice. In the open field test, *Arid1b*$^{+/-}$ mice spent significantly more time in the periphery while

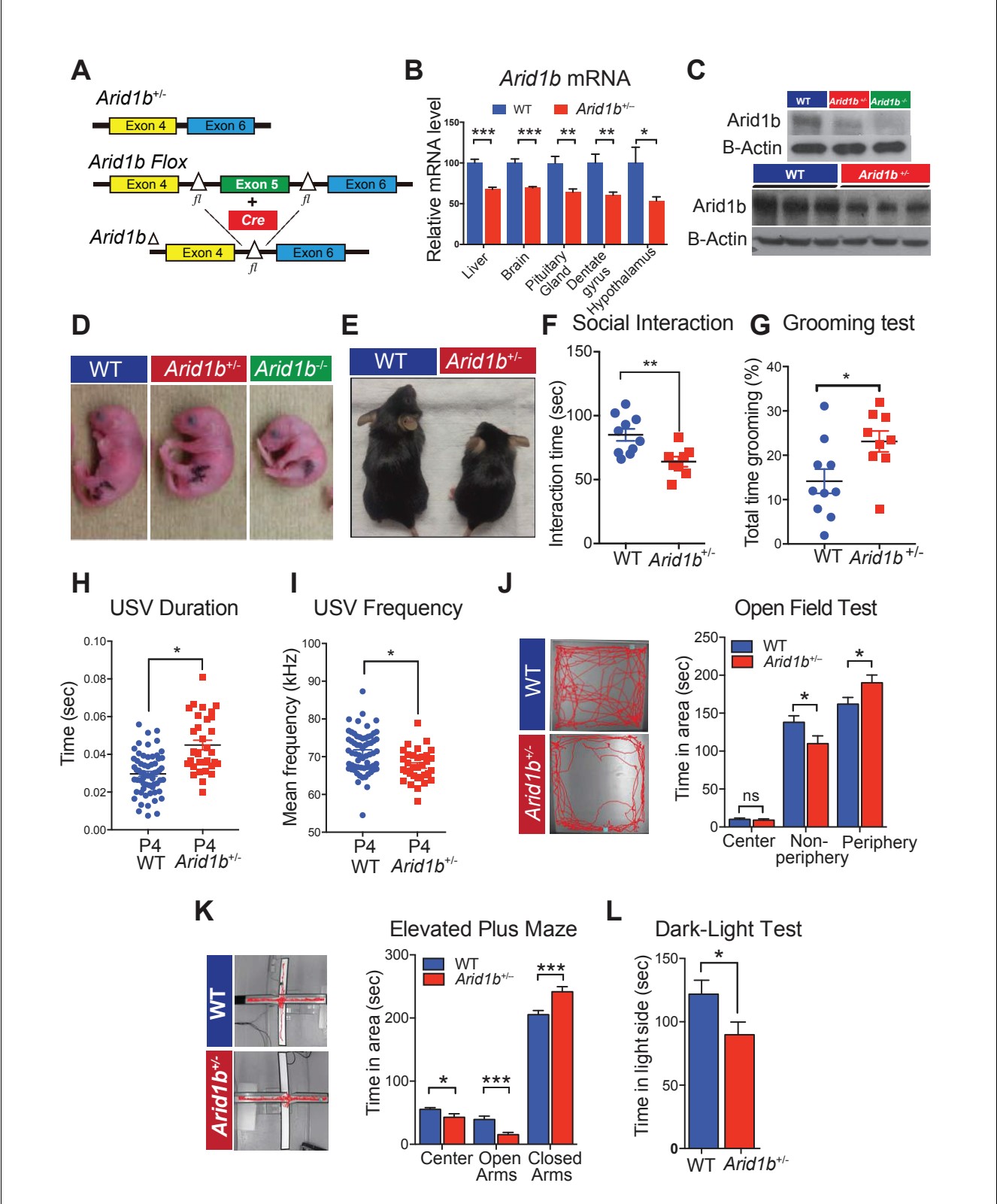

**Figure 1.** *Arid1b*[+/-] mice exhibit physical manifestations of developmental delay, autistic-like features, and abnormal behavioral phenotypes. (**A**) Schematic of *Arid1b* whole body heterozygous mice in which exon five is deleted (hereafter referred as *Arid1b*[+/-]) and *Arid1b* floxed mice. (**B**) Relative *Arid1b* mRNA levels in selected organs and brain regions as assessed by qPCR. (**C**) Relative Arid1b levels in p0 mouse limb (top panel) and whole brain extracts at p45 as assessed by western blot analysis. (**D**) Appearance of WT and *Arid1b*[+/-] mice at postnatal day 0. (**E**) Appearance of WT and *Arid1b*[+/-]

*Figure 1 continued on next page*

*Figure 1 continued*

littermates at 1 month of age. (F) Juvenile social interaction testing for 10 WT and 9 *Arid1b*$^{+/-}$ male mice. (G) Grooming test for 10 WT and 9 *Arid1b*$^{+/-}$ female mice. (H, I) The ultrasonic vocalization (USV) test measuring the duration and frequency of vocal communication in 63 WT and 33 *Arid1b*$^{+/-}$ male and female mice during separation of pups from dams at postnatal day 4. (J) Representative traces of WT and *Arid1b*$^{+/-}$ mice in the open field and time spent in the indicated areas for 20 WT and 20 *Arid1b*$^{+/-}$ 8 week old male mice. (K) Representative traces of WT and *Arid1b*$^{+/-}$ mice in the elevated plus maze and time spent in the indicated areas for 20 WT and 20 *Arid1b*$^{+/-}$ 8 week old male mice. (L) Dark-light box testing for 20 WT and 20 *Arid1b*$^{+/-}$ 8 week old male mice. Values represent mean ± SEM. Asterisks indicate significant differences between indicated littermate genotypes, *p-value ≤ 0.05; **p-value ≤ 0.01; ***p-value ≤ 0.001; ****p-value ≤ 0.0001; ns, not significant. Student's *t*-test (two-tailed distribution, two-sample unequal variance) was used to calculate p-values unless otherwise indicated in the figure legend.

The following source data and figure supplements are available for figure 1:

**Source data 1.**

**Figure supplement 1.** Additional characterization of *Arid1b*$^{+/-}$ mice.

**Figure supplement 1—source data 1.**

**Figure supplement 2.** Additional neurobehavioral testing on *Arid1b*$^{+/-}$ mice.

**Figure supplement 2—source data 1.**

avoiding the anxiety-provoking center (*Figure 1J*). In the elevated plus maze, *Arid1b*$^{+/-}$ mice spent more time in the anxiety-relieving, walled arms of the maze (*Figure 1K*). In the dark-light box test, *Arid1b*$^{+/-}$ mice avoided exploring the brightly lit chamber (*Figure 1L*). WT and mutant mice traveled equal distances both initially and over a 2 hr time period, making locomotor differences less likely a confounder in simple environments (*Figure 1—figure supplement 2D*). These tests consistently demonstrated higher levels of anxiety-like behavior in *Arid1b*$^{+/-}$ mice compared to their WT littermates.

Given the associations between *Arid1b* haploinsufficiency and intellectual disability, we assessed cognitive functions in *Arid1b*$^{+/-}$ mice. The Morris water maze test, a contextual fear-conditioning test, and a cued fear-conditioning test each did not reveal defects in memory and learning (*Figure 1—figure supplement 2E–G*). The genotypes were equally able to sense the electric shock applied during fear conditioning (*Figure 1—figure supplement 2H*). Overall, these tests showed that *Arid1b*$^{+/-}$ mice displayed abnormal social, vocal, and behavioral phenotypes, but did not clearly have cognitive or memory deficiencies.

## *Arid1b* haploinsufficiency resulted in neuroanatomical and gene expression abnormalities

In an effort to understand how behavioral abnormalities arose, we examined *Arid1b*$^{+/-}$ brains for other neurodevelopmental abnormalities. Because some patients with *ARID1B* mutations exhibit corpus callosum hypoplasia or agenesis (*Schrier Vergano et al., 2013*), we examined brains of *Arid1b*$^{+/-}$ mice and identified a significant reduction in corpus callosum volume (*Figure 2A*). Consistent with studies showing that small hippocampus, dentate gyrus, and cortex size are associated with anxiety and depressive disorders in mice and humans (*Persson et al., 2014*; *Travis et al., 2015*; *Boldrini et al., 2013*; *Schmaal et al., 2017*), *Arid1b*$^{+/-}$ mice have smaller dentate gyri (*Figure 2B*) and both *Arid1b*$^{+/-}$ and *Arid1b*$^{-/-}$ pups had reduced cortical thickness with reduced TBR1 marked neuronal cellularity (*Figure 2—figure supplement 1A–D*). Less proliferating cells were also seen in the subgranular zone of the dentate gyrus (*Figure 2C,D,F,G*), especially in posterior regions (*Figure 2E,H*). Thus, reduced corpus callosum size, dentate gyrus size, cortex thickness, and proliferation are neuroanatomical and cellular correlates of the behavioral phenotypes seen in *Arid1b* mutants.

RNA-seq was performed to examine the impact of *Arid1b* haploinsufficiency on transcriptional output in the hippocampus. Differential gene expression analysis showed 56 significantly down- and 79 upregulated mRNAs (edgeR FDR < 0.05; *Figure 3A*). As expected, *Arid1b* was one of the most

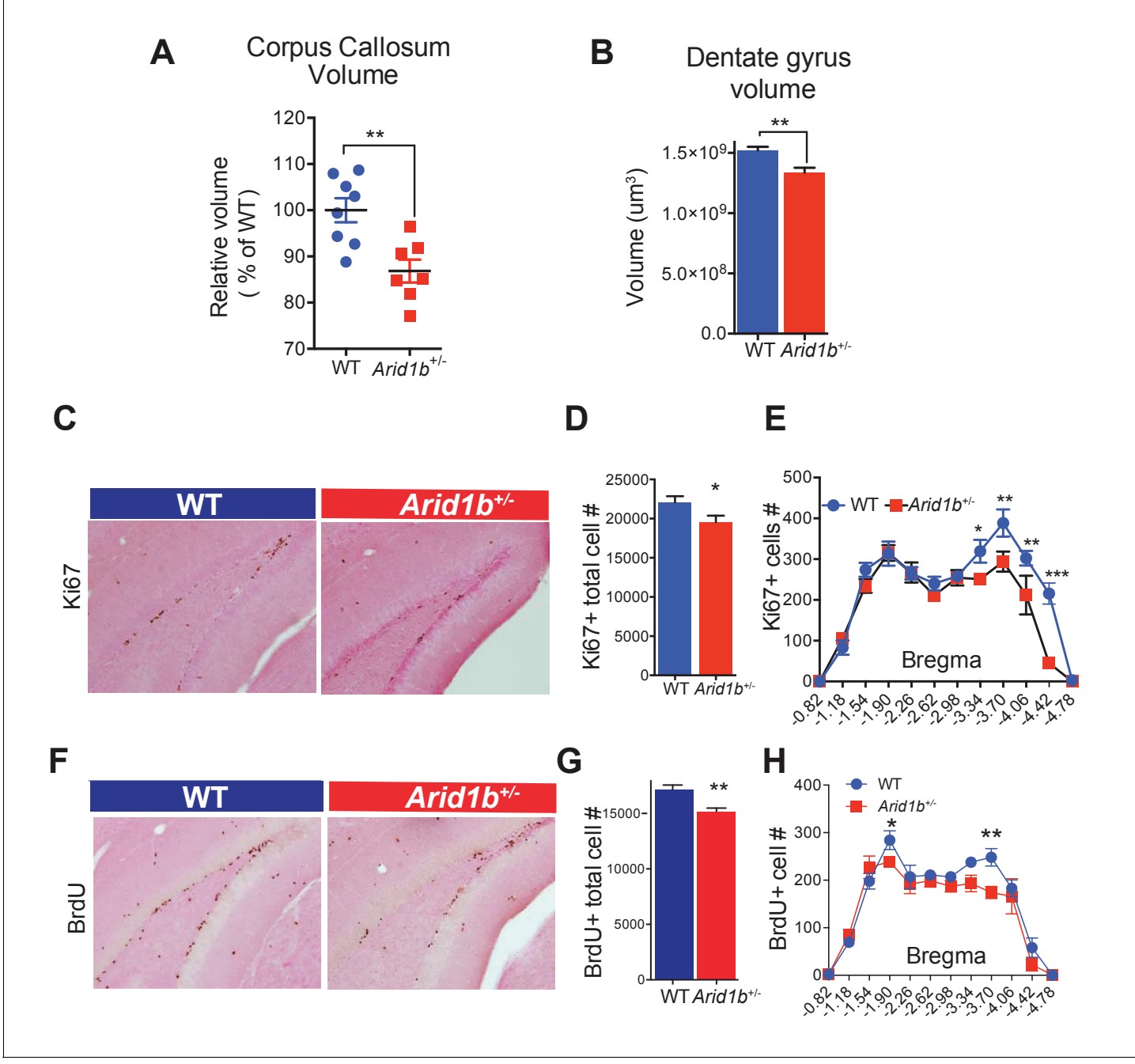

**Figure 2.** *Arid1b* haploinsufficiency results in neuroanatomical abnormalities implicated in neuropsychiatric diseases. (**A**) Relative corpus callosum volume quantified through Cavalieri analysis (n = 8 WT and 7 *Arid1b*[+/-] brains from 50 day old females). (**B**) Dentate gyrus volume quantified through Cavalieri analysis (n = 7 WT and 7 *Arid1b*[+/-] brains from 50 day old females). (**C**) Representative Ki67 immunostaining. (**D**) Quantitation of Ki67+ total cell number (8 WT and 7 *Arid1b*[+/-] brains from 50 day old females). (**E**) Bregma analysis was used to determine cell proliferation (Ki67) as a function of location in the subgranular zone of the dentate gyrus. Two-way ANOVA with uncorrected Fischer's Least Significant Difference (LSD) was used to calculate the statistics. (**F**) Representative BrdU immunostaining. WT and *Arid1b*[+/-] mice received one injection per day of the thymidine analog, bromodeoxyuridine (BrdU), for five days and brains were harvested three days following the last injection (6 WT and 4 *Arid1b*[+/-] brains from 50 day old females). (**G**) Quantification of BrdU+ total cell number. (**H**) Bregma analysis was used to determine cell proliferation (BrdU) as a function of location in the subgranular zone of the dentate gyrus (n = 6 WT and 4 *Arid1b*[+/-]). Values represent mean ± SEM. Asterisks indicate significant differences between indicated littermate genotypes, *p-value $\leq$ 0.05; **p-value $\leq$ 0.01; ***p-value $\leq$ 0.001; ****p-value $\leq$ 0.0001; ns, not significant. Student's *t*-test (two-tailed distribution, two-sample unequal variance) was used to calculate p-values unless otherwise indicated in the figure legend.

The following source data and figure supplements are available for figure 2:

*Figure 2 continued on next page*

*Figure 2 continued*

Source data 1.
Figure supplement 1. *Arid1b⁺/⁻* and *Arid1b⁻/⁻* brains have defects in cortical development.
Figure supplement 1—source data 1.

downregulated genes. Globally, differentially regulated genes were associated with nervous system development as well as psychological, behavioral, and developmental disorders (*Figure 3B*). *Arid1b⁺/⁻* tissues also showed specific alterations in Ephrin, nNOS, axonal guidance and glutamate receptor signaling pathways (*Figure 3C*). 14 of 140 (10%) differentially regulated genes were among the highest ranking candidate autism risk genes identified in the SFARI gene database (*Basu et al., 2009*) (*Figure 3D*). To determine if some of these genes could be directly regulated by SWI/SNF, we analyzed the ChIP-Seq targets of Smarca4 (Brg1), a core SWI/SNF complex subunit (*Attanasio et al., 2014*). 91 of 140 (65%) differentially regulated genes showed direct binding by Brg1 (*Figure 3E*), with positional enrichment at transcriptional start sites (TSSs) (*Figure 3F,G*). Arid1b-mediated SWI/SNF transcriptional activities appeared to directly regulate numerous neuropsychiatric related genes, including ones implicated in ASD (*Figure 3H,I*). Our data also show that haploinsufficiency is sufficient to cause broad gene expression disruption, but future studies will be required to determine the exact downstream genes that account for the neuropsychiatric phenotypes.

## *Arid1b⁺/⁻* mice exhibited GHRH-GH-IGF1 axis defects

Having establishing that *Arid1b* haploinsufficient mice recapitulate salient aspects of human *ARID1B*-opathies, we were particularly interested in identifying reversible pathological mechanisms and therapeutic opportunities. Since we identified neuroanatomical and neural expression aberrations in *Arid1b⁺/⁻* mice, we also asked if any non-neuropsychiatric syndromic features were potentially related to neurodevelopmental abnormalities. As mentioned previously, *Arid1b⁺/⁻* mice developed reduced nose-to-rump length and weight (*Figure 4A,B*). *Arid1b⁺/⁻* mice had disproportionally small kidneys and hearts, but no other gross organ defects (*Figure 4—figure supplement 1A*). Profiling using metabolic cages showed that *Arid1b⁺/⁻* mice had equivalent food intake and water consumption (*Figure 4—figure supplement 1B,C*), suggesting that size differences were unlikely due to food intake or energetic differences.

A critical factor that regulates both body size and brain development is Insulin-like growth factor (IGF1). We found a significant reduction in plasma IGF1 levels in *Arid1b⁺/⁻* mice (*Figure 4C*), and confirmed a reduction in *Igf1* mRNA in the liver, which is a major source of IGF1 (*Figure 4D*). To discern if there was a hypothalamic, pituitary, peripheral, or combinatorial problem that led to IGF1 deficiency, we performed a series of endocrinologic tests. In the same cohorts of mice with IGF1 deficiency, fasting GH was not significantly different in *Arid1b⁺/⁻* mice (*Figure 4E*). In addition, there were no significant *Gh* mRNA expression differences between WT and *Arid1b⁺/⁻* pituitary glands despite differences in *Arid1b* mRNA levels (*Figure 4F*). We also confirmed that plasma GH was not altered in younger 2 week old *Arid1b⁺/⁻* mice, an age where GH levels are more critical for growth (*Figure 4G*). The combination of low IGF1 and normal GH levels pointed to a peripheral defect without appropriate GH compensation from the pituitary gland.

Because GH was not elevated as would be expected if there was only a peripheral IGF1 producing defect, we asked if the *Arid1b⁺/⁻* pituitary was capable of making and secreting sufficient amounts of GH in the context of GH stimulation testing. In multiple cohorts of *Arid1b⁺/⁻* mice, GH levels were never significantly different at baseline and also increased normally at multiple time points after stimulation with Growth hormone-releasing hormone (GHRH) (*Figure 4H*). This indicated a normal ability for the pituitary to respond to exogenous GHRH. Next, we attempted to determine if the hypothalamus was not producing enough *Ghrh*. We dissected the mediobasal hypothalamus containing *Ghrh* expressing neurons to examine *Ghrh* expression. We found that *Ghrh* mRNA levels were not significantly different (*Figure 4I*), indicating a lack of appropriate GHRH response to IGF1 deficiency, suggesting a central defect that contributed to growth impairment.

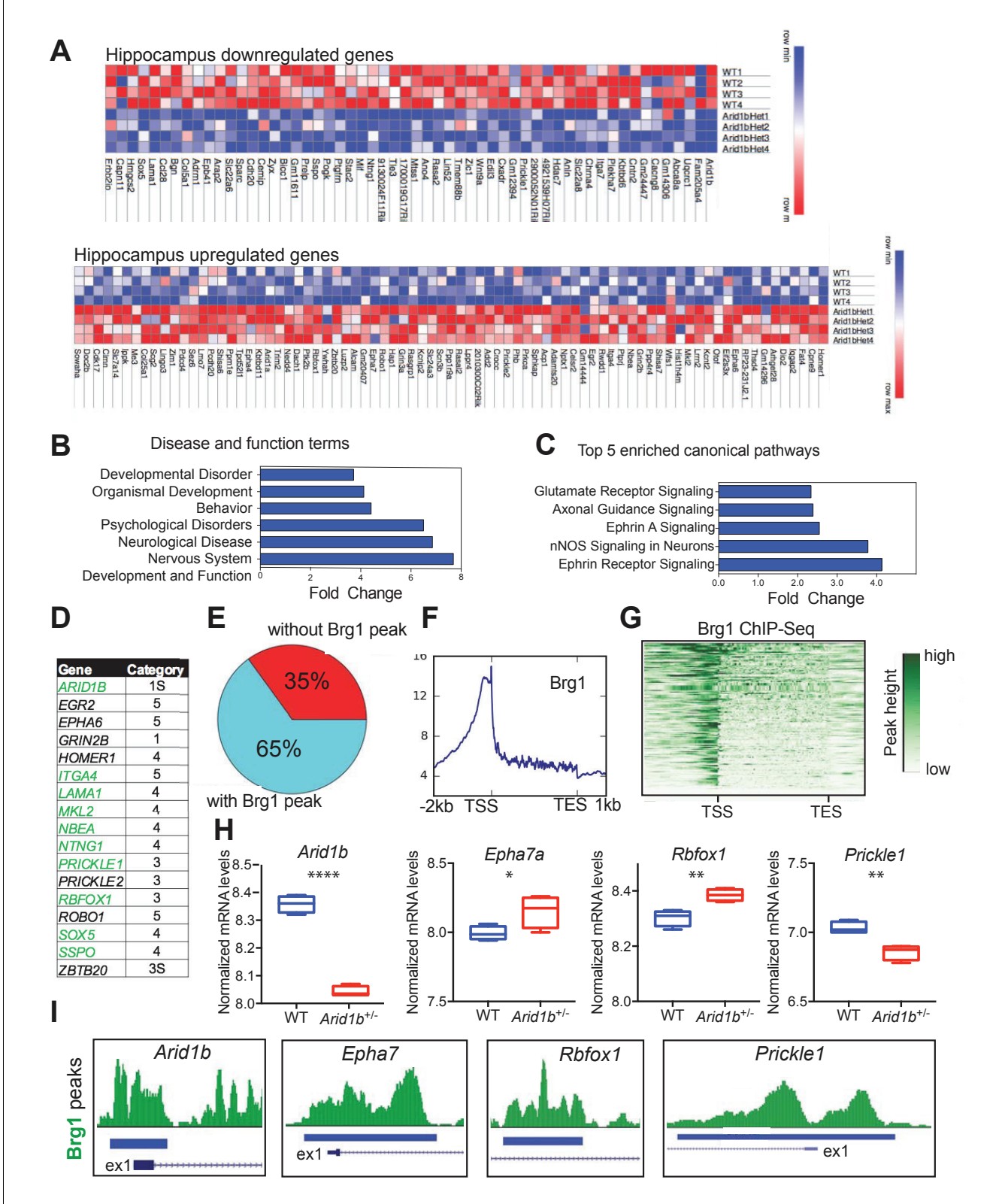

**Figure 3.** *Arid1b* haploinsufficiency results in changes in the expression of SWI/SNF regulated genes implicated in neuropsychiatric diseases. (**A**) All significantly up- and downregulated genes in the *Arid1b*[+/-] hippocampus are ranked according to p-value (least to most significant from left to right). (**B**) Most enriched diseases and biological functions in hippocampus. (**C**) Most differentially regulated pathways in the hippocampus. (**D**) 14 of 140 (10%) differentially regulated genes were among the highest ranking autism risk genes identified in the SFARI database. Category S: syndromic, Category 1:

*Figure 3 continued on next page*

*Figure 3 continued*

high confidence, Category 2: strong candidate, Category 3: suggestive evidence, Category 4: minimal evidence, Category 5: Hypothesized (*Basu et al., 2009*). (E) Pie chart showing that 91 of 140 (65%) differentially regulated genes in hippocampus are direct targets of Brg1, a core SWI/SNF complex subunit. Brg1 target genes were identified using ChIP-Seq in mouse e11.5 forebrain (*Attanasio et al., 2014*). (F) Metaplot showing enrichment of Brg1 at the TSSs of genes regulated by Arid1b. (G) Heatmap showing Brg1 promoter binding in these genes. (H) Differential mRNA expression of representative genes involved in neurodevelopment and ASD (Data from: SFARI database, updated September, 2016) (*Basu et al., 2009*). (I) Brg1 peaks suggesting direct binding of SWI/SNF at the promoters of ASD-related genes. Values represent mean ± SEM. Asterisks indicate significant differences between indicated littermate genotypes, *p-value ≤ 0.05; **p-value ≤ 0.01; ***p-value ≤ 0.001; ****p-value ≤ 0.0001; ns, not significant. Student's *t*-test (two-tailed distribution, two-sample unequal variance) was used to calculate p-values unless otherwise indicated in the corresponding figure legend.

In addition, we sought genetic evidence for a partial central (hypothalamic or pituitary) root cause of IGF1 deficiency by generating organ-specific *Arid1b* mutant models. We used *Nestin-Cre* in the brain, *Albumin-Cre* in the liver, and *Ckmm-Cre* in the skeletal muscles to spatially control *Arid1b* haploinsufficiency. Neither *Albumin-Cre; Arid1b$^{Fl/+}$* (*Figure 4J*) nor *Ckmm-Cre; Arid1b$^{Fl/+}$* mice (data not shown) showed growth or morphological defects, while *Nestin-Cre; Arid1b$^{Fl/+}$* mice recapitulated the growth impairments seen in whole body *Arid1b$^{+/-}$* mice (*Figure 4K*), suggesting at least a neuronal contribution to growth impairment. While *Albumin-Cre; Arid1b$^{Fl/+}$* mice showed no IGF1 differences, *Nestin-Cre; Arid1b$^{Fl/+}$* mice had reduced plasma IGF1 levels (*Figure 4L,M*). Moreover, *Nestin-Cre; Arid1b$^{Fl/+}$* mice showed an inappropriate lack of GH increase in the face of this IGF1 deficiency (*Figure 4N*). Since liver specific *Arid1b$^{+/-}$* mice did not replicate the whole body *Arid1b$^{+/-}$* mice, it is possible that a combination of central and multi-organ peripheral defects in the GHRH-GH-IGF1 axis were required to fully recapitulate the growth impairment of whole body *Arid1b$^{+/-}$* mice.

## GH therapy reversed growth retardation and muscle weakness

Given plasma IGF1 deficiency in *Arid1b$^{+/-}$* cohorts, we first tested if IGF1 replacement could rescue physical aspects of developmental delay and abnormal behavioral phenotypes. Neither body size (*Figure 5A*) nor elevated plus maze abnormalities (*Figure 5B*) were rescued after treating WT and *Arid1b$^{+/-}$* cohorts with recombinant human IGF1 (rhIGF1). This was not surprising because it is known that exogenous IGF1 is unstable and often does not efficiently reach target tissues responsible for growth (*Kletzl et al., 2017*).

The fact that GH was not elevated in the context of low IGF1 suggested to us that there was not adequate GH production or compensation. Thus, we asked if GH supplementation could rescue some of the physical aspects of developmental delay. WT and *Arid1b$^{+/-}$* cohorts were treated with recombinant mouse GH (rmGH) (*Figure 5C*). After 40 days of treatment, *Arid1b* heterozygous mice gained significantly more body weight and nose-to-rump length than did WT mice (*Figure 5D,E*), demonstrating that exogenous GH supplementation was sufficient to rescue growth retardation in *Arid1b$^{+/-}$* mice. Given this selective efficacy for mutant mice, we asked if GH could potentially improve muscle weakness often associated with CSS. We found that at baseline, *Arid1b$^{+/-}$* mice also had muscle weakness identified through grip strength testing. Replacement with GH was able to selectively increase muscle strength in mutant mice (*Figure 5F,G*). Despite improvements in physical manifestations, GH replacement was not able to reverse behavioral phenotypes such as anxiety, as measured in the elevated plus maze (*Figure 5H*). This suggested that correcting the GHRH-GH-IGF1 axis was not sufficient to rescue neuropsychiatric manifestations, but was able to reverse growth retardation mediated by *Arid1b* deficiency.

In an analysis of 60 *ARID1B* CSS patients, height was shown to be significantly reduced (*Santen et al., 2014*). In addition, some non-syndromic patients with missense mutations in *ARID1B* exhibited growth deficiency due to partial GH deficiency (*Yu et al., 2015*). We also obtained clinical information from additional CSS patients, two with *ARID1B* mutations (from the www.arid1bgene.com database) and one with a mutation in *SMARCA4*, which encodes another SWI/SNF component. All three of these cases had deficiencies in the GH-IGF1 axis and clear beneficial responses to GH replacement therapy (growth curves for the *ARID1B* patients are shown in *Figure 5—figure*

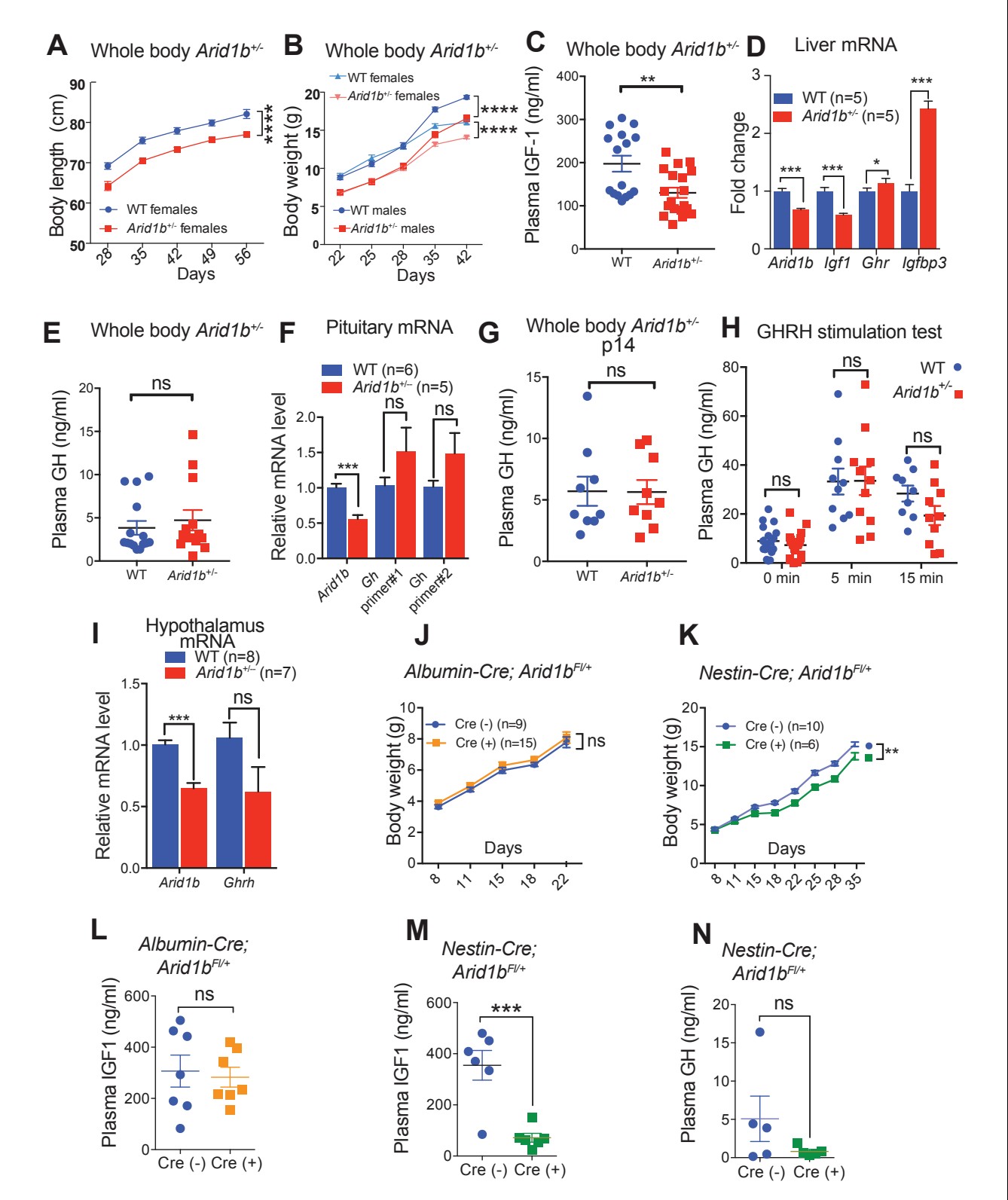

**Figure 4.** Growth retardation in *Arid1b*[+/−] mice is due to GH-IGF1 axis deficiency with a neuronal source. (**A**) Body length (nose-to-rump) curve of females (n = 9 WT and 9 *Arid1b*[+/−]). (**B**) Body weight growth curve for males (n = 14 WT and 14 *Arid1b*[+/−]) and females (n = 20 WT and 20 *Arid1b*[+/−]). For (**A**) and (**B**), repeated ANOVA with Bonferroni's post-hoc analysis was used. (**C**) Plasma IGF1 as measured by ELISA (n = 16 WT and 19 *Arid1b*[+/−] 28–41 day old male and female mice). (**D**) *Igf1* mRNA in WT and *Arid1b*[+/−] livers as measured by qPCR (n = 5 WT and 5 *Arid1b*[+/−] livers from 45 day old female

*Figure 4 continued on next page*

*Figure 4 continued*

mice). (E) Plasma GH as measured by ELISA (n = 15 WT and 14 *Arid1b*<sup>+/-</sup> 28–41 day old male and female mice). (F) *Gh* mRNA in WT and *Arid1b*<sup>+/-</sup> pituitary as measured by qPCR (n = 6 WT and 5 *Arid1b*<sup>+/-</sup> pituitary from 33-44 day old female mice). (G) Plasma GH (n = 9 WT and 9 *Arid1b*<sup>+/-</sup> 2 week old male mice). (H) Plasma GH before and after stimulation by human GHRH (n = 19 WT and n = 20 *Arid1b*<sup>+/-</sup> mice at baseline, n = 11 WT and n = 10 *Arid1b*<sup>+/-</sup> mice 5 and 15 min after GHRH administration). (I) *Ghrh* mRNA in WT and *Arid1b*<sup>+/-</sup> mediobasal hypothalamus as measured by qPCR. (n = 8 WT and 7 *Arid1b*<sup>+/-</sup> samples from 33-44 day old female mice). (J) Body weight curve for female *Arid1b*<sup>Fl/+</sup> (n = 9) and *Albumin-Cre; Arid1b*<sup>Fl/+</sup> (n = 15) mice. (K) Body weight curve for female *Arid1b*<sup>Fl/+</sup> (n = 10) and *Nestin-Cre; Arid1b*<sup>Fl/+</sup> (n = 6) mice. (L) Plasma IGF1 levels for 40–45 day old female *Arid1b*<sup>Fl/+</sup> (n = 7) and *Albumin-Cre; Arid1b*<sup>Fl/+</sup> (n = 7) mice. (M) Plasma IGF1 levels for 30–45 days old female *Arid1b*<sup>Fl/+</sup> (n = 6) and *Nestin-Cre; Arid1b*<sup>Fl/+</sup> (n = 6) mice. (N) Plasma GH levels for female *Arid1b*<sup>Fl/+</sup> (n = 5) and *Nestin-Cre; Arid1b*<sup>Fl/+</sup> (n = 5) mice. Values represent mean ± SEM. Asterisks indicate significant differences between indicated littermate genotypes, *p-value $\leq$ 0.05; **p-value $\leq$ 0.01; ***p-value $\leq$ 0.001; ****p-value $\leq$ 0.0001; ns, not significant. Student's *t*-test (two-tailed distribution, two-sample unequal variance) was used to calculate p-values unless otherwise indicated in the figure legend.

The following source data and figure supplements are available for figure 4:

**Source data 1.**
**Figure supplement 1.** Growth and metabolic analysis of *Arid1b*<sup>+/-</sup> mice.
**Figure supplement 1—source data 1.**

supplement 1A,B). These data from humans and mice suggest deficiencies at various parts of the GHRH-GH-IGF axis, leading to growth impairment responsive to GH supplementation.

## Discussion

In summary, we have developed the first mouse model of *Arid1b* haploinsufficiency, one of the most common genetic lesions found in ASD, ID, and CSS. Several aspects of our model recapitulated the features of the overlapping disorders associated with *ARID1B* (*Table 1*). Our model is faithful to the heterozygosity seen in most *ARID1B*-opathies. It is interesting that another model of ASD involving a chromatin remodeling gene called *CHD8* also requires haploinsufficiency (*Katayama et al., 2016*) and suggests that dose could play a critical mechanistic role in phenotypes resulting from mutations in epigenetic regulators. Our study also raises interesting questions about how genotype relates to phenotype in diseases involving *ARID1B*. Some children with *ARID1B* mutations have a subset but not all features of CSS, and others have different disorders such as ASD or isolated corpus callosum agenesis. For example, Yu *et al.* reported cases of idiopathic short stature without cognitive defects that were attributed to de novo *ARID1B* missense mutations, suggesting that there are either differences between mutations or important genetic interactions that play a key role in defining the phenotypic expression of these mutations (*Yu et al., 2015*). It is also possible that differences between the function of *Arid1b* in mouse and human brain development account for intellectual and cognitive discrepancies. Our mouse model affords the ability to interrogate these types of questions in different strain backgrounds and with genetic interactors. We are hopeful that this will advance the understanding of *ARID1B* and SWI/SNF in human diseases.

We also uncovered a role for the GHRH-GH-IGF1 axis in *ARID1B*-opathies. Previously, height was shown to be reduced in *ARID1B* patients (*Santen et al., 2014*), but there are only anecdotal findings of GH deficiencies (*Figure 5—figure supplement 1A,B*) (*Yu et al., 2015*). After the clinical identification of *ARID1B* or SWI/SNF mutations, interventions for short stature are usually not investigated. Thus, it is likely that GHRH-GH-IGF1 deficiency is under-diagnosed and rarely treated in this patient population. Conversely, *ARID1B* mutations are not suspected in patients with non-syndromic short stature, and could represent a more common causative mechanism than previously suspected. The findings here should motivate deeper interrogation of the GHRH-GH-IGF1 axis and potentially GH supplementation in syndromic patients with CSS or non-syndromic patients with *ARID1B* mutations and short stature.

Our study does not pinpoint the exact source of the peripheral IGF1 deficiency since liver-specific Cre lines did not recapitulate the IGF1 deficiency seen in the whole body *Arid1b*<sup>+/-</sup> mice. Given the *Nestin-Cre* results, it is possible that reduced IGF1 production in the brain led to reduced plasma

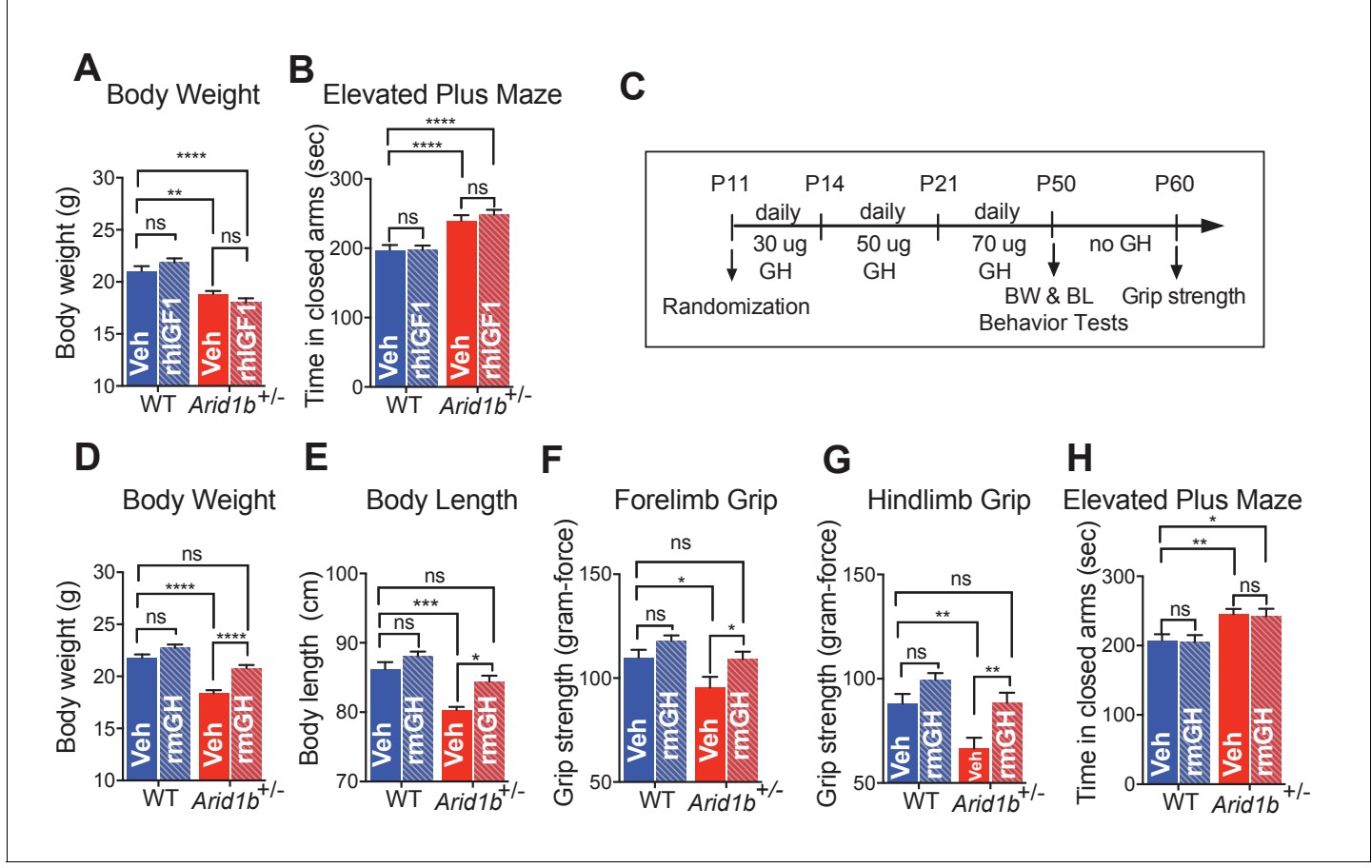

**Figure 5.** GH therapy reverses growth retardation and muscle weakness. (**A**) Body weights at p50 (WT + vehicle (n = 12), WT + rhIGF1 (n = 12), *Arid1b*[+/-] + vehicle (n = 13) and *Arid1b*[+/-] + rhIGF1 (n = 13)). (**B**) Time spent in the closed arms of elevated plus maze at p50 (WT + vehicle (n = 27), WT + rhIGF1 (n = 29), *Arid1b*[+/-] + vehicle (n = 21) and *Arid1b*[+/-] + rhIGF1 (n = 22)). For (**A**) and (**B**), 0.5 mg/kg rhIGF1 was administrated daily starting from postnatal day 11. (**C**) Schema showing the duration and dose of daily recombinant GH treatment. (**D**) Body weights at p50 (WT + vehicle (n = 20), WT + rmGH (n = 16), *Arid1b*[+/-] + vehicle (n = 16) and *Arid1b*[+/-] + rmGH (n = 16)). (**E**) Nose-to-rump lengths at p50 (WT + vehicle (n = 7), WT + rmGH (n = 7), *Arid1b*[+/-] + vehicle (n = 6) and *Arid1b*[+/-] + rmGH (n = 6)). (**F**) Forelimb grip strength at p50 (WT + vehicle (n = 9), WT + rmGH (n = 9), *Arid1b*[+/-] + vehicle (n = 9) and *Arid1b*[+/-] + rmGH (n = 9)). (**G**) Hindlimb grip strength at p50 (WT + vehicle (n = 9), WT + rmGH (n = 9), *Arid1b*[+/-] + vehicle (n = 9) and *Arid1b*[+/-] + rmGH (n = 9)). (**H**) Time spent in the closed arms of elevated plus maze at p50 (WT + vehicle (n = 19), WT + rmGH (n = 16), *Arid1b*[+/-] + vehicle (n = 19) and *Arid1b*[+/-] + rmGH (n = 16)). Values represent mean ± SEM. Asterisks indicate significant differences between indicated littermate genotypes: *p-value ≤ 0.05; **p-value ≤ 0.01; ***p-value ≤ 0.001; ****p-value ≤ 0.0001; ns, not significant. Two-way ANOVA was used to calculate the p-value.

The following source data and figure supplement are available for figure 5:

**Source data 1.**

**Figure supplement 1.** Two *ARID1B* mutant CSS patients were GH deficient and responsive to GH replacement.

IGF1 and the inability to compensate with GHRH and GH exacerbated growth retardation. Another possibility that subtle peripheral defects in the liver and muscle will only manifest when combined with defects in other organs such as the brain. Future studies with tissue-specific conditional experiments could help to resolve these questions. Overall, our study provides a preclinical model for mechanistic and therapeutic dissection of *ARID1B* related diseases, and offers a translatable avenue to alleviate growth related aspects of developmental delay.

**Table 1.** Major clinical features associated with *ARID1B* mutations and phenotypes seen in *Arid1b*[+/-] mice. Abbreviations: CSS: Coffin-siris syndrome, ID: Intellectual disability, ASD: Autism spectrum disorder.

| Human ARID1B features | Diagnosis | References | Mouse |
|---|---|---|---|
| Intellectual/cognitive disability | CSS, ID | (*Hoyer et al., 2012*; *Santen et al., 2012*; *Halgren et al., 2012*) | No |
| Growth retardation | CSS | (*Tsurusaki et al., 2012*) | Yes |
| Coarse facial features | CSS | (*Sim et al., 2015*) | Unknown |
| Muscle hypotonia | CSS | (*Hoyer et al., 2012*) | Yes |
| Hydrocephalus | CSS | (*Imai et al., 2001*) | Yes |
| Corpus callosum agenesis or hypoplasia | CSS | (*Halgren et al., 2012*; *Santen et al., 2012*) | Yes |
| Brachydactyly, hypoplastic nail/finger | CSS | (*Hoyer et al., 2012*; *Santen et al., 2012*; *Brautbar et al., 2009*) | Unknown |
| Abnormal vocalization, speech impairment | ASD, CSS | (*Santen et al., 2012*) | Yes |
| Anxiety | ASD, CSS | (*O'Roak et al., 2012*; *Deciphering Developmental Disorders Study, 2015*) | Yes |
| Social interaction deficits | ASD | (*O'Roak et al., 2012*; *Deciphering Developmental Disorders Study, 2015*) | Yes |
| Repetitive behaviors | ASD | (*O'Roak et al., 2012*; *Deciphering Developmental Disorders Study, 2015*) | Yes |

# Materials and methods

## Mice

All animal procedures were based on animal care guidelines approved by the Institutional Animal Care and Use Committee at University of Texas Southwestern Medical Center (UTSW). Constitutive and conditional *Arid1b* knockout mice were generated by the UTSW Transgenic Core using CRISPR/Cas9 genome editing. Guide RNAs were designed to target sequences before and after exon 5 of *Arid1b*, creating a frame-shift mutation to induce nonsense-mediated decay. Guide RNAs, S. pyogenes *Cas9* mRNA, and oligo donors containing LoxP sequences were injected into single celled zygotes. C57BL/6J mice were used to generate these mice. To generate WT and *Arid1b*[+/-] study mice, C57BL/6J WT females were crossed to *Arid1b*[+/-] males. *Arid1b*[+/-] mice were tail genotyped using the primers CTTGGTCTTACCCATTTGCACAGT (forward) and GATGGAGGATCCTTACTACAGGGGGATT (reverse). Amplicon size for WT allele is 710 bp and deletion band is 310 bp. *Arid1b* floxed mice were tail genotyped using the primers 5'-CTT GGT CTT ACC CAT TTG CAC AGT-3' (forward) and 5'-AGT GCC TAG GAA GGC AGA GTT TGA GAG-3' (reverse). Amplicon size for WT allele is 475 bp, and for floxed allele is 554 bp.

## Pituitary and hypothalamus dissection

The whole pituitary and the mediobasal hypothalamus (MBH), which includes both the arcuate nucleus (ARC) and the ventromedial (VMH) hypothalamus, were dissected and subjected to RT-qPCR analysis.

## RNA extraction and qRT-PCR

Total RNA was isolated using TRIzol reagent (Invitrogen (Carlsbad, California), catalog no. 15596018). cDNA synthesis was performed with 1 µg of total RNA using the iScript RT Supermix (BioRad, catalog no. 1708840). SYBR Green based quantitative real-time PCR was performed. Gene expression levels were measured using the $\Delta\Delta$Ct method. Mouse *Arid1b* primers were: forward, 5'-GTTGGCTCTCCTGTGGGAAGCAA-3'; reverse, 5'-GTGACTGGCTCAAGGCAGGAT-3'.

## Western blot assay

Tissues were lysed in TPER tissue protein extraction reagent and homogenized in FastPrep tissue homogenizer. Western blots were performed in standard fashion. Primary antibodies were prepared in 5% BSA in PBS-T. The following primary antibody was used: Arid1b Antibody (KMN1) X (cat #: sc-32762 X, RRID:AB_2060367). Following secondary antibody was used: anti-mouse IgG, HRP-linked antibody (Cell Signaling, #7076, RRID:AB_330924).

## Immunohistochemistry (IHC)

Perinatal mice were decapitated after anesthesia. Heads were fixed in 4% PFA overnight before brains were extracted. Adult mice were anesthetized and underwent intracardial perfusion with ice-cold 0.1 M phosphate-buffered saline (PBS) followed by 4% PFA for fixation. Extracted brains were immersed for 24 hr in 4% PFA in 0.1M PBS at 4°C for post-fixation, followed by least three days of immersion in 30% sucrose in 0.1 M PBS with 0.01% sodium azide for cryoprotection. Brains were cut into 40 μm- (adult) or 20 μm-thick (perinatal) sections with a cryostat (model CM3050S; Leica). Brain sections were permeabilized with 0.25% Triton X-100 in 1×PBS and then blocked for 2 hr with 5% BSA/3% normal goat serum (NGS) in 0.25% Triton X-100 in 1×PBS. Primary antibodies for TBR1 (1:150 dilution, rabbit, polyclonal, Abcam Cat# ab31940 RRID:AB_2200219) were applied to sections overnight at 4°C. To count cell number, brain sections were incubated with Hoechst 33342 (1 μg/ml; Cell Signaling Technology Cat# 4082S RRID:AB_10626776) alone or together with secondary antibodies (Alexa 488) for 2 hr at room temperature.

For the analysis of proliferation, adult mice received one BrdU injection for five consecutive days, and three days following the last injection, brains were fixed and harvested. Slide-mounted IHC for BrdU-, Ki67-, and doublecortin- immunoreactive (+) cells in dentate gyrus was performed as described previously (*DeCarolis et al., 2014*; *Walker et al., 2015*). Briefly, every ninth section of the hippocampus was slide-mounted onto charged slides and left for two hours to dry. Antigen retrieval was performed using 0.01 M citric acid (pH 6.0) at 100°C for 15 min, followed by washing in PBS at room temperature (RT). Hydrogen peroxide (0.3% $H_2O_2$) incubation was performed for 30 min to inhibit endogenous peroxidase activity. For BrdU IHC, permeabilization with 0.1% Trypsin in 0.1 M TRIS and 0.1% $CaCl_2$ and denaturation with 2N HCl in 1x PBS were performed in order to allow antibody access to nuclear DNA. One hour of blocking non-specific binding was performed by incubation in 3% donkey serum, 0.3% Triton-X in PBS. Following these steps, slides were incubated with rat-α-BrdU (1:400; Bio-Rad / AbD Serotec Cat# OBT0030 RRID:AB_609568), rabbit-α-Ki67 antibody (1:500; Thermo Fisher Scientific Cat# RM-9106-S0 RRID:AB_2341197) in 3% serum and 0.3% Tween-20 overnight. Primary antibody incubation was followed by 1 x PBS rinses and a 1 hr incubation with a biotin-tagged secondary antibody targeting the respective primary antibody. Following rinses with 1x PBS, incubation with an avidin-biotin complex occurred for 90 min. Incubation with diaminobenzidine for 5–10 min was used to visualize immunoreactive cells. The counterstain Fast Red was used for nuclear visualization (~3 min incubation). Lastly, slides were placed through an ethanol dehydration series and coverslipped with DPX.

## Stereological cell counts

BrdU+ and Ki67+ cells were quantified using an Olympus BX-51 microscope at 40X by an observer blind to experimental groups as previously described (*Walker et al., 2015*). Immunopositive cells were quantified in every 9th coronal section in the subgranular zone of the granular cell layer in the dentate gyrus, spanning the entire anterior-posterior axis of the hippocampus (−0.82 mm to −4.78 mm from Bregma). Manual stereological counting was performed under bright field, and total cell counts were multiplied by nine to account for the whole hippocampus. Doublecortin+ cells were quantified using the Optical Fractionator method, and dentate gyrus and corpus callosum volume were assessed using Cavalieri analysis. Stereoinvestigator was used to perform both of these techniques. Investigators were blinded to genotypes during sectioning, counting, and analysis.

## RNA-Seq and ChIP-Seq

RNA from four WT and four *Arid1b*$^{+/-}$ hippocampi from 78 to 82 day old females were purified with a QIAGEN miRNeasy Mini Kit. NuGEN libraries were made with these RNAs. These indexed libraries were multiplexed in a single flow cell lane and received 75 base single-end sequencing on a NextSeq 500 using the High Output Kit v2 (75 cycles) at the CRI Sequencing Facility. Raw sequencing reads were trimmed to remove adaptor and low quality sequences (Phred score <20) using trim_galore package (http://www.bioinformatics.babraham.ac.uk/projects/trim_galore/). Trimmed reads were aligned to mouse reference genome GRCm38/mm10 with HiSAT2 *Kim et al., 2015*. After duplicates removal by SAMtools (*Li et al., 2009*) and Picard (http://broadinstitute.github.io/picard.), read counts were generated for the annotated genes based GENCODE V20 (*Harrow et al., 2012*) using featureCounts (*Liao et al., 2014*). Differential gene analysis was performed use

edgeR (*Robinson et al., 2010*), using FDR < 0.05 as cutoff. Enriched pathways were analyzed through the use of QIAGEN's Ingenuity Pathway Analysis (IPA, QIAGEN RedwoodCity, www.qiagen. com/ingenuity). Heatmaps to visualize the data were generated by using GENE-E (www.broadinstitute.org/cancer/software/GENE-E). RNA-Seq data is deposited to GEO database and can be accessed through GEO accession number (GSE92238). Brg1 ChIP-seq data from the forebrain was downloaded from GEO with the accession number GSM912547 in GSE37151. Files were remapped to the mm10 genome build by CrossMap. Brg1 target genes were predicted by using BETA-minus program on the Cistrome Analysis Pipeline, an integrative platform for transcriptional regulation studies. Heatmap and Metaplot were generated using deeptools, a flexible platform for exploring deep-sequencing data.

## Metabolic cage studies

Metabolic cage studies were performed in a temperature-controlled room containing 36 TSE metabolic cages (The TSE Labmaster System of Germany) maintained by UTSW Metabolic Core. Three days prior to study, mice were introduced to metabolic cages and after three day acclimation, cages were connected to TSE system and parameters were recorded for a total of five days. Investigators were blinded to mouse genotypes.

## Grip strength test

Muscle strength was measured by a grip strength test performed by the Neuro-Models Core Facility at UT Southwestern Medical Center in a blinded fashion. Test was conducted using a wire mesh grid connected to a horizontally-aligned force meter (San Diego Instruments, San Diego, CA). The grid was secured at a 45 degree angle, and the top rung of the grid was used for all testing. Mice were held at the base of the tail and supported ventrally while being moved into position to grasp the wire grid. Once the rug was successfully grasped, mice were gently pulled in a horizontal plane until the animal's grip was released from the grid. Peak force (in gram-force units, gf) was captured by the force meter and recorded for later analysis. Forelimb and hindlimb tests were conducted separately, with each being measured five times over a 2–3 min period. Investigators were blinded to mouse genotypes and the identity of treatment groups. *Arid1b*$^{+/-}$ mice with apparent hydrocephaly were excluded. Experiment was performed once.

## Ultrasonic vocalization recordings

Ultrasonic vocalizations (USVs) were recorded from both male and female pups isolated from their mothers at P4, during the daylight period of the light/dark cycle. Dams and their litters were acclimated for 30 min in the test room. Each pup was removed from the cage containing its mother and littermates and placed in a clean plastic container in a wooden sound-attenuating recording chamber. Each pup was first acclimated in the recording chamber for 30 s then recorded for 10 min. Recordings were acquired using an UltraSoundGate CM16/CMPA condenser microphone (Avisoft Bioacoustics) positioned at a fixed height of 8 cm above the pups, and were amplified and digitized (sampled at 16 bits, 250 kHz) using UltraSoundGate 416 hr 1.1 hardware and Avisoft-RECORDER software (Avisoft Bioacoustics). The data were transferred to Avisoft-SASLab Pro (version 5.2) to analyze spectrograms of vocalizations with settings of 0% overlapping FlatTop windows, 100% frame size, and 256 points fast Fourier transform (FFT) length. The following measures were recorded for each group: number of USV calls, mean duration of USV calls, and mean peak frequency. Recordings were performed with the experimenter blinded to mouse genotypes.

## Juvenile social interaction test

The adult male test mouse was placed into a fresh home cage and habituated in the test room with red light for 15 min before testing. A three week old male juvenile mouse was placed into the opposite end of the cage that the test mouse was already in. Active interactions between the mice were scored manually with 2 min of total test time. Only interactions when the test mouse is interacting with the juvenile, but not other way around were scored. Non-strict male littermates were used. *Arid1b*$^{+/-}$ mice with obvious hydrocephaly were excluded. The experiment was performed twice.

## Grooming test

The test was performed between 10:00 am and 2:00 pm. Adult female mouse was singly placed into a new standard cage, without nestlets, food, or water, acclimated for 10 min, then videotaped for another 10 min. The amount of time spent grooming was recorded continuously to calculate the total time spent grooming. Grooming is considered self-grooming of any part of the body (including the face, head, ears, full-body). Data is plotted as percent of total time spent grooming.

## Marble burying test

The test mouse was acclimated for 30 min in the testing room. One standard housing cage for each test mouse was filled with clean bedding material. 15 clean marbles were arranged on top of the bedding in each cage, forming five even rows and three columns. Mice were placed individually into the prepared cages and kept undisturbed for 30 min. After the testing period, they were returned to their original cages. A still image of the test cage was taken to record the number and pattern of buried marbles. A marble was considered buried if more than 2/3 of its depth is covered. Results were calculated and plotted as the percentage of marbles buried per genotype.

## Locomotor activity

This test was performed by UTSW Rodent Core Facilty. Experimenters were blinded to mouse genotypes. The experiment was repeated two times and combined data is included. Mice were placed individually into a clean, plastic mouse cage (18 cm x 28 cm) with minimal bedding. Each cage was placed into a dark Plexiglas box. Movement was monitored by 5 photobeams in one dimension (Photobeam Activity System, San Diego Instruments, San Diego, CA) for 2 hr, with the number of beam breaks recorded every 5 min. The movement is characterized in three ways: repetitive beam breaks of a single beam is classified as stereotypy, consecutive beam breaks of two or more beams is classified as ambulatory movements, and total beam breaks during each 5 min interval. Number of total beam breaks during 5 min interval was reported. $Arid1b^{+/-}$ mice with apparent hydrocephaly were excluded.

## Open field activity test

The test was performed by UTSW Rodent Core Facilty. Experimenters were blinded to mouse genotypes. Experiment was repeated two times and combined data is included. Mice were placed in the periphery of a novel open field environment (44 cm x 44 cm, walls 30 cm high) in a dimly lit room and allowed to explore for 5 min. The animals were monitored from above by a video camera connected to a computer running video tracking software (Ethovision 3.0, Noldus, Leesburg, Virginia) to determine the time, distance moved and number of entries into three areas: the periphery (5 cm from the walls), the center (14 cm x 14 cm) and non-periphery (area excluding periphery). The open field arenas were wiped and allowed to dry between mice. Locomotor activity test was performed prior to open field activity test in these cohorts. $Arid1b^{+/-}$ mice with apparent hydrocephaly were excluded.

## Elevated plus maze

Test was performed by UTSW Rodent Core Facilty. Experimenters were blinded to mouse genotypes. Experiment was repeated two times and combined data is included. Mice were placed in the center of a black Plexiglas elevated plus maze (each arm 30 cm long and 5 cm wide with two opposite arms closed by 25 cm high walls) elevated 31 cm in a dimly lit room and allowed to explore for 5 min. The animals were monitored from above by a video camera connected to a computer running video tracking software (Ethovision 3.0, Noldus, Leesburg, Virginia) to determine time spent in the open and closed arms, time spent in the middle, and the number of entries into the open and closed arm. The apparatus was wiped and allowed to dry between mice. Locomotor activity and open field activity test were performed prior to elevated plus maze in these cohorts. $Arid1b^{+/-}$ mice with apparent hydrocephaly were excluded.

## Dark-Light activity

Test was performed by UTSW Rodent Core Facilty. Experimenters were blinded to mouse genotypes. Experiment was repeated two times and combined data is included. Mice were placed into a

black Plexiglas chamber (25 cm x 26 cm) and allowed to explore for 2 min. After the habituation period, a small door was opened allowing them to access the light side of the apparatus (25 cm x 26 cm lit to approximately 1700 lux) for 10 min. The animals were monitored by 7 photobeams in the dark compartment and 8 photobeams on the light side connected to a computer which recorded the time spent in each compartment, latency to enter the light side and the number of entrances to each compartment (Med-PC IV, Med Associates, St. Albans, VT). The dark-light apparatus was wiped and allowed to dry between mice. Locomotor activity, open field activity, and elevated plus maze tests were performed prior to dark-light activity test in these cohorts. *Arid1b*$^{+/-}$ mice with apparent hydrocephaly were excluded.

## Morris water maze

Test was performed by UTSW Rodent Core Facilty. Experimenters were blinded to mouse genotypes. A circular pool was filled with room temperature water to a depth of approximately 12 inches. A platform (10 cm diameter) was placed in one quadrant of the pool with the top of the platform about 2 cm below the water level. White non-toxic paint was added to enhance the contrast with the animal and to hide the location of the platform. Each day the mice were placed in the pool and allowed to swim for 1 min to find the platform. The swim path and time until locating the platform wasrecorded via a videocamera and computer running videotracking software (Ethovision, Noldus). If the mouse does not find the platform within a minute, they were gently guided or placed on the platform for 10 s, then removed from the pool and return to their home cage. Each animal was placed in the pool for a total of four times each day for 13 days. Twenty-four hours after the last training day, a probe test was conducted in which the platform was removed from the pool and each mouse was allowed to swim for 1 min to determine whether the animal has learned the location of the platform. The time each animal spends in the quadrant which had contained the platform on training days and the number of times that the animal crosses the location which had contained the platform are indicators of how well the animal has learned the spatial location of the platform. To control for visual problems, the mice weregiven 4–6 trials after the probe test using the same pool and platform, however a large black block was placed on top of the platform to clearly mark the location. The location of the platform was moved on each trial.

## Fear conditioning

Test was performed by UTSW Rodent Core Facilty. Experimenters were blinded to mouse genotypes. Fear conditioning was measured in boxes equipped with a metal grid floor connected to a scrambled shock generator (Med Associates Inc., St. Albans). For training, mice were individually placed in the chamber. After 2 min, the mice received three tone-shock pairings (30 s white noise, 80 dB tone co-terminated with a 2 s, 0.5 mA footshock, 1 min intertrial interval). The following day, memory of the context was measured by placing the mice into the same chambers and freezing was measured automatically by the Med Associates software. Forty-eight hours after training, memory for the white noise cue was measured by placing the mice in a box with altered floors and walls, different lighting, and a vanilla smell. Freezing was measured for 3 min, then the noise cue was turned on for an additional 3 min and freezing was measured.

## Footshock sensitivity

Test was performed by the UTSW Rodent Core Facility. Experimenters were blinded to mouse genotypes. The mice were placed individually into boxes equipped with a metal grid floor connected to a scrambled shock generator (Med Associates Inc., St. Albans). After approximately 1 min, the mice received a series of footshocks (2 s each) with increasing intensity. The initial shock intensity was 0.05 mA and the amplitude was increased by 0.05 mA for each consecutive footshock with 15 s intershock interval. The first shock intensity that each animal displayed each behaviour (flinch, jump, and vocalization) was recorded. Once the animal displayed all three behaviours, it was removed from the chamber.

## GH stimulation test and ELISA experiments

Plasma IGF-1 concentration was determined using mouse/rat IGF-1 Quantikine ELISA Kit (R and D Biosystems, Cat #: MG100) without fasting. Mice were fasted for 36 hr before GH stimulation

testing. Fifteen minutes after anesthesia with pentobarbital (50 mg/kg given once i.p.), 0.14 g/kg GHRH (Phoenix Pharmaceuticals, Cat #:031–02) was injected i.p. Blood was sampled retro-orbitally using a capillary tube before, 5, and 15 min after injection. Plasma GH concentration was measured using a Rat/Mouse GH ELISA KIT (Millipore, Cat #: EZRMGH-45K).

## Recombinant human IGF1 (rhIGF1) therapy

See *Figure 5C*. Mice at the age of P10 were ranked from highest to lowest body weight and even numbered mice were placed into the treatment group and odd numbered mice were placed into the vehicle group. *Arid1b*$^{+/-}$ mice with apparent hydrocephaly were excluded. Recombinant human IGF1 was purchased from Peprotech (Catalog:#100–11). It was prepared according to datasheet and injected intraperitoneally. Vehicle or 0.5 mg/kg rhIGF1 dissolved in vehicle were injected starting from P11 daily.

## Recombinant mouse growth hormone (rmGH) therapy

Mice at the age of P11 were ranked from highest to lowest body weight and even numbered mice were placed into the treatment group and odd numbered mice were placed into the vehicle group. *Arid1b*$^{+/-}$ mice with apparent hydrocephaly were excluded. Recombinant mouse growth hormone was obtained from National Hormone and Peptide Program (NHPP). It was prepared according to the NHPP datasheet. Injection was performed subcutaneously. 30 ug GH/mouse/day was injected between P11 to P14, 50 ug GH/mouse/day was injected between P14 to P21, 70 ug GH/mouse/day was injected between P21 to P50. Grip strength test was performed at P60 after 10 days without treatment.

## Statistical analyses

Unless specified otherwise in the figure legends, statistical analyses were performed using unpaired, two-tailed, Student's t-test or ANOVA. The data bars and error bars indicate mean ± standard error mean (SEM). *p-value $\leq$ 0.05; **p-value $\leq$ 0.01; ***p-value $\leq$ 0.001; ****p-value $\leq$ 0.0001; ns, not significant. No statistical methods were used to predetermine sample sizes; however, sample sizes were estimated based on similar experiments reported in the relevant literature in the field (*Araujo et al., 2015* and *Katayama et al., 2016*).

# Acknowledgements

We thank Jiang Wu, Eric Olson, and Jian Xu for critical input and advice. We thank Dr. Eric Olson's lab for providing *Ckmm-Cre* mice. We thank the Children's Research Institute Sequencing Core for sequencing and UTSW Bioinformatics Core for the analysis. We thank Dr. Erik Plautz and Laura Ingle in the UTSW Neuro-Models Facility for performing the grip strength test and UTSW Rodent Behavior Core Facility for performing behavioral tests.

# Additional information

### Funding

| Funder | Grant reference number | Author |
| --- | --- | --- |
| Hamon Center for Regenerative Science and Medicine | Regenerative medicine training grant | Cemre Celen Xuxu Sun |
| Postdoctoral Institutional training grant | NIDA T32-DA007290 | Angela K Walker |
| HHMI International Student Research Fellowship | | Liem H Nguyen |
| National Institutes of Health | DA023701 | Amelia J Eisch |
| National Institutes of Health | DA023555 | Amelia J Eisch |
| National Institutes of Health | MH107945 | Amelia J Eisch |
| National Institute of Neurological Disorders and Stroke | NINDS K99/R00 (R00NS073735) | Woo-Ping Ge |

| National Institutes of Health | R21(NS099950) | Woo-Ping Ge |
| --- | --- | --- |
| Pollock Foundation | | Hao Zhu |
| National Institutes of Health | 1K08CA157727 | Hao Zhu |
| National Cancer Institute | 1R01CA190525 | Hao Zhu |
| Burroughs Wellcome Fund | | Hao Zhu |
| CPRIT New Investigator Award | R1209 | Hao Zhu |
| CPRIT Early Translation Grant | DP150077 | Hao Zhu |
| University of Texas Southwestern Medical Center | | Maria Chahrour |
| NARSAD Young Investigator Grant | NARSAD 24951 | Maria Chahrour |
| Cancer Prevention and Research Institute of Texas | RP150596 | Xin Luo |
| Barts Health NHS Trust | | Evelien F Gevers |
| Queen Mary University London | | Evelien F Gevers |

The funders had no role in study design, data collection and interpretation, or the decision to submit the work for publication.

## Author contributions

CC, Conceptualization, Resources, Software, Supervision, Funding acquisition, Validation, Investigation, Visualization, Methodology, Writing—original draft, Project administration, Writing—review and editing; J-CC, XL, Conceptualization, Data curation, Formal analysis, Supervision, Validation, Investigation, Visualization, Methodology, Writing—original draft, Writing—review and editing; NN, Formal analysis, Validation, Visualization, Methodology, Writing—review and editing; AKW, Data curation, Formal analysis, Methodology; FC, Data curation, Formal analysis, Validation, Methodology, Writing—review and editing; SZ, GWES, Data curation, Formal analysis, Visualization, Methodology; ASC, IN, AB, XS, LAB, MM, EFG, SGB, Data curation; LHN, Data curation, Formal analysis, Methodology, Writing—review and editing; AJE, Conceptualization, Data curation, Formal analysis, Supervision, Validation, Investigation, Methodology; CMP, Conceptualization, Supervision, Validation, Investigation, Methodology, Writing—review and editing; W-PG, Conceptualization, Writing—review and editing; MC, Conceptualization, Data curation, Investigation, Writing—review and editing; HZ, Conceptualization, Resources, Data curation, Formal analysis, Supervision, Validation, Investigation, Methodology, Writing—review and editing

## Author ORCIDs

Cemre Celen, http://orcid.org/0000-0001-8308-3750
Hao Zhu, http://orcid.org/0000-0002-8417-9698

## Ethics

Human subjects: Patient data included in the article is non-identifiable data, and hence does not require approval from the patients.
Animal experimentation: All animal procedures were based on animal care guidelines approved by the Institutional Animal Care and Use Committee at University of Texas Southwestern Medical Center (UTSW). Animal protocol number is 2015-101118.

# Additional files

## Major datasets

The following dataset was generated:

| Author(s) | Year | Dataset title | Dataset URL | Database, license, and accessibility information |
|---|---|---|---|---|
| Celen C, Chuang J, Luo X, Zhu H | 2016 | RNA-seq transcriptonal profiling in whole hippocampus in WT and Arid1b+/- mice | https://www.ncbi.nlm.nih.gov/geo/query/acc.cgi?acc=GSE92238 | Publicly available at the NCBI Gene Expression Omnibus (accession no: GSE92238) |

The following previously published datasets were used:

| Author(s) | Year | Dataset title | Dataset URL | Database, license, and accessibility information |
|---|---|---|---|---|
| Blow MJ, Nord AS, Pennacchio LA, Attanasio C | 2014 | mouse_e11.5_forebrain_smarca4-flag | https://www.ncbi.nlm.nih.gov/geo/query/acc.cgi?acc=GSM912547 | vailable at the NCBI Gene Expression Omnibus (accession no: GSM912547) |
| Raab JR, Resnick S, Magnuson T | 2015 | Genome-Wide Transcriptional Regulation Mediated By Biochemically Distinct Forms of SWI/SNF | https://www.ncbi.nlm.nih.gov/geo/query/acc.cgi?acc=GSE69568 | Publicly available at the NCBI Gene Expression Omnibus (accession no: GSE69568). |

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
