## [Decision Letter]

Thank you for submitting your article "*Arid1b* haploinsufficient mice reveal neuropsychiatric phenotypes and reversible causes of growth impairment" for consideration by *eLife*. Your article has been reviewed by two peer reviewers, and the evaluation has been overseen by Joseph Gleeson as the Reviewing Editor and Huda Zoghbi as the Senior Editor. The reviewers have opted to remain anonymous.

The reviewers have discussed the reviews with one another and the Reviewing Editor has drafted this decision to help you prepare a revised submission.

Summary

The authors generate a null and conditional Arid1b heterozygous mouse model, and test it for features related to human ARID1B-haploinsufficiency disorders and Coffin-Siris syndrome. They note growth retardation, small corpus callosum, and altered behavior in models of anxiety. They did not note deficiencies in learning or memory that would be expected from the clinical finding in patients of intellectual disability. Finally, they observed rescue of growth and grip strength with administration of growth hormone.

This is a compelling manuscript, the first to demonstrate neurobehavioral deficits associated with Arid1b haploinsufficiency, the first to demonstrate a CNS-root of the growth retardation, and the first to and suggest GH replacement as a treatment. Strong points of this paper are the multiple behavioral testing paradigms presented by authors suggesting that Arid1b heterozygosity correlates with a neuropsychiatric phenotypes. There are a couple leaps in logic that make the correlation of this model with the human phenotypes feel at times forced, but overall the results are interesting. Despite these shortcomings, (i.e. IGF1 is low, but does not rescue phenotype. GH is not convincingly low, but does rescue the phenotype.) the findings are presented in a compelling way.

Concerns

Subsection “*Arid1b^+/-^*mice exhibit central GH-IGF1 axis deficiency”. The serum GH sampling reduction does not appear to be statistically significant, and as well appears to be driven by a single outlier high measurement of a normal sample. Please increase numbers and remove statistical outliers. If the authors are implicating that it is inappropriately low in the context of low IGF1, it would be important to show this in the same serum sample, and would be useful to demonstrate a similar effect in a more proven model of GH insufficiency.

The GH deficiency model is interesting, and because of the clinical relevance, should be strengthened. For example, GH levels may be difficult to directly test accurately. To prove a GH deficiency in these mice, it may be useful to utilize GH stimulation test (please clarify what test precisely you think authors should perform). Igfbp3 reduction may also be supportive of GH deficiency.

CSS is about 4x more common in females than males. Did the authors find any evidence to suggest the reason for this discrepancy that could relate to pituitary function, sex hormone alterations, or tolerance of mutation in males? Are there any difference in phenotypes between male and female Arid1b heterozygous and KO mice? Many tests were only done in male or female but not in both. In selection of genders for each test, are there any criteria that authors applied?

Reviewer #1:

The authors report an Arid1b heterozygous mouse model, and test it for features related to human ARID1B-haploinsufficiency disorders. They note small stature, and suggest GH deficiency as an underlying cause, small corpus callosum, and altered behavior in models of anxiety. They did not note deficiencies in learning or memory. Finally, they are able to observe rescue of growth and grip strength with administration of growth hormone.

The work is presented well with some minor typographical errors. There are a couple leaps in logic that make the correlation of this model with the human phenotypes feel at times forced, but overall the results are interesting.

My two main areas of concern relate to

Did they test for GH deficiency and/or correct it in patients?

Some items feel overstated and should downgrade the wording from "proven" to "possible" findings or provide specific data to clearly support the hypothesis or statement.

Specific instances:

Introduction section. "Our findings not only functionally validate ARID1B's involvement in human disease, they also reveal underappreciated clinical manifestations of human ARID1B mutations that can be corrected in the clinic." Would perhaps better be worded "Our findings not only functionally validate ARID1B's involvement in human disease, they suggest underappreciated clinical manifestations of human ARID1B mutations that could be corrected in the clinic."

Subsection “*Arid1b^+/-^*mice develop abnormal social, vocal, and behavioral phenotypes”. Autism is not a heavily reported or observed feature of CSS/ARID1B patients, the most common group of patients reported. They may be largely because autism can be difficult to ascertain in the context of intellectual disability. It may be useful to provide a statement in the Introduction noting the specific data that support this correlation.

Regarding some of the mouse findings:

Subsection “*Arid1b^+/-^*mice exhibit developmental delay, hydrocephalus, and digit abnormalities”. The difference in frequency of polydactyly/brachydactyly in Arid1 +/- mice from controls statistically significant (p=0.07). This information isn't directly related to the focus, so I would recommend removing it.

Subsection “*Arid1b^+/-^*mice develop abnormal social, vocal, and behavioral phenotypes”. Are USV duration and pitch correlated with ASD behavior? This may be useful to state explicitly.

Subsection “*Arid1b^+/-^*mice exhibit central GH-IGF1 axis deficiency”. The serum GH sampling reduction does not appear to be statistically significant, and as well appears to be driven by a single outlier high measurement of a normal sample. If the authors are implicating that it is inappropriately low in the context of low IGF1, it would be important to show this in the same serum sample, and would be useful to demonstrate a similar effect in a more proven model of GH insufficiency.

Loose connections…IGF1 is low, but does not rescue phenotype. GH is not convincingly low, but does rescue the phenotype.

Discussion section. Santen et al., 2014, demonstrate modest growth impairment, but from what I can read in that paper, they did not remark on growth hormone deficiency as noted in the text.

The growth hormone model is interesting, and because of the clinical relevance, should be clarified and strengthened. For example, GH levels may be difficult to directly test accurately. To prove a GH deficiency in these mice, it may be useful to utilize a growth hormone stimulation test. Igfbp3 reduction may also be supportive of GH deficiency.

As this proposed model of GH insufficiency is a key element of this work, I would emphasize slightly more the often underappreciated/noted small stature in these patients (Santen et al., 2014) in the introduction and abstract.

Discussion points:

The lack of memory and learning defects is quite notable. Can the authors elaborate on potential reasons for this?

Reviewer #2:

Zhu and colleagues modeled ARID1B-opathies in mouse. Strong point of this paper are the multiple behavioral evidence presented by authors suggesting that Arid1b heterozygosity caused neuropsychiatric phenotypes. Overall, the paper is well-written and figures/data are presented in high quality. Although haploinsufficiency in the components of mammalian SWI/SNF complex is now well-known cause of neuropsychiatric disorders, the novelty lies in the reversal of growth defect by GH treatment.

Using the conditional KO animals, the authors speculated that the peripheral IGF1 deficiency due to low GH level is the underlying cause of short stature:

1) What is the primary pituitary defects of Arid1b heterozygous mouse?

2) How pituitary (defect) causes IGF1 deficiency in Arid1b heterozygote?

As authors mentioned, some experiments were not applicable to strengthen major conclusions:

3) no effect of rhIGH1 treatment due to low plasma stability

4) no detection of GH deficiency in "hippocampal" RNA-seq

Paper could be improved by testing the key speculations using simple methods.

---

## [Author Response]

Summary

*The authors generate a null and conditional Arid1b heterozygous mouse model, and test it for features related to human ARID1B-haploinsufficiency disorders and Coffin-Siris syndrome. They note growth retardation, small corpus callosum, and altered behavior in models of anxiety. They did not note deficiencies in learning or memory that would be expected from the clinical finding in patients of intellectual disability. Finally, they observed rescue of growth and grip strength with administration of growth hormone.*

*This is a compelling manuscript, the first to demonstrate neurobehavioral deficits associated with Arid1b haploinsufficiency, the first to demonstrate a CNS-root of the growth retardation, and the first to and suggest GH replacement as a treatment. Strong points of this paper are the multiple behavioral testing paradigms presented by authors suggesting that Arid1b heterozygosity correlates with a neuropsychiatric phenotypes. There are a couple leaps in logic that make the correlation of this model with the human phenotypes feel at times forced, but overall the results are interesting. Despite these shortcomings, (i.e. IGF1 is low, but does not rescue phenotype. GH is not convincingly low, but does rescue the phenotype.) the findings are presented in a compelling way.*

Thank you for this feedback. Before we address specific questions, the following two paragraphs describe data that was not requested but which we think would help strengthen the conclusions of the paper. To enrich the connections between *Arid1b*^+/-^ mice and ASD-like phenotypes, we performed grooming and marble burying tests that examine repetitive behaviors (*[42]*). Consistent with other ASD mouse models, *Arid1b*^+/-^ mice exhibited increased grooming (Figure 1) and potentially as a consequence, buried less marbles (Figure 1—figure supplement 2). Similar repetitive behaviors were seen with *Synapsin* knockout mice, which exhibit features of ASD (*[27]*). It is unclear why *Arid1b*^+/-^ mice bury marbles less, but one possibility is that grooming time is significantly increased, thus reducing time spent on burying. These data strengthen the functional connection between *Arid1b* mutations and ASD.

We also examined cellularity and cortical thickness of brains from WT, *Arid1b^+/-^,* and *Arid1b^-/-^* P1 pups. This analysis showed that cortical thickness and cellularity declined with loss of *Arid1b* alleles (Figure 2—figure supplement 1 to D). TBR1, an early neuronal marker, showed that reduced cortical thickness was accompanied by reduced neuron numbers. These data are in line with the reduced hippocampal and dentate gyrus size, and suggest global changes in brain cellularity.

*Concerns*

*Subsection “Arid1b^+/-^mice exhibit central GH-IGF1 axis deficiency”. The serum GH sampling reduction does not appear to be statistically significant, and as well appears to be driven by a single outlier high measurement of a normal sample. Please increase numbers and remove statistical outliers. If the authors are implicating that it is inappropriately low in the context of low IGF1, it would be important to show this in the same serum sample, and would be useful to demonstrate a similar effect in a more proven model of GH insufficiency.*

*The GH deficiency model is interesting, and because of the clinical relevance, should be strengthened. For example, GH levels may be difficult to directly test accurately. To prove a GH deficiency in these mice, it may be useful to utilize GH stimulation test (please clarify what test precisely you think authors should perform). Igfbp3 reduction may also be supportive of GH deficiency.*

We agree that the GH-IGF1 hormone measurements could be more robust and probed with greater numbers and depth. Thus, we performed a series of experiments with large N’s to try to identify where the GHRH-GH-IGF1 axis is defective in *Arid1b*^+/-^ mice. First, we confirmed the plasma IGF1 deficiency in additional cohorts. The original cohort is on the left (n = 7 WT and 6 HET; 45 day old female mice). The new cohorts are on the right (n = 16 WT and 19 HET, 21-33 day old male and female mice). This new data was added to the paper as Figure 4.

In more simplistic cases such as dwarfism from IGF1 mutations, Laron dwarfism (mutant GH receptor), or pituitary tumors causing GH deficiencies, the source of the IGF1 deficiency is usually clear. For example, children with Laron dwarfism do not respond to GH treatment due to a lack of GH receptors. In cases where there are whole-body syndromic defects that affect multiple organs, this source can often be more difficult to pinpoint. For example, it is not entirely clear what the source of IGF1 deficiency is in Rett Syndrome, despite the ability for IGF1 to rescue components of the disorder in mice (*[9]*).

In an attempt to pinpoint the hypothalamic, pituitary, peripheral, or combinatorial problem that was leading to the IGF1 deficiency, we performed a series of endocrinological and genetic tests. In the same cohorts of mice with IGF1 deficiency mentioned above (Figure 4), GH was not significantly different in the *Arid1b*^+/-^ mice in fed or fasting conditions (Figure 4 in the paper revision). We also confirmed that GH was not altered in younger 2-week old *Arid1b^+/-^* mice, an age where GH levels are more critical for growth (Figure 4). This suggested a peripheral defect without appropriate compensation from either the pituitary or hypothalamus.

Examination of four additional cohorts shown below also showed no significant differences in plasma GH levels in either fed or fasting states. These data were not included in the paper (Figure 6).

Author response image 1.**DOI:**
http://dx.doi.org/10.7554/eLife.25730.022

To determine if the pituitary is capable of making and secreting sufficient amounts of GH, we performed GH stimulation testing with Growth Hormone Releasing Hormone (GHRH) (*[12]*). In two independent experiments with different cohorts of mice, we determined that in *Arid1b*^+/-^ mice, GH levels were never significantly different at baseline and also increased normally at multiple time points after stimulation (Figure 4). In addition, GH mRNA levels in the pituitary as measured by qPCR were not significantly changed (Figure 4). In sum, this clearly indicated a normal ability for the pituitary to respond to exogenous GHRH.

Next, we attempted to determine if the hypothalamus is not producing enough GHRH. To examine the expression of endogenous GHRH, we dissected the mediobasal hypothalamus (MBH), which includes both the arcuate nucleus (ARC) and the ventromedial (VMH) hypothalamus, two regions that normally produces GHRH. We found the expected reduction of *Arid1b* mRNA, but no significant increase in GHRH mRNA (n = 8 WT and 7 HET female mice) (Figure 4). The lack of upregulation by GHRH in response to low IGF1 suggested a partial deficiency at the level of the hypothalamus. The fact that GHRH and GH were not significantly lower strongly supports the existence of a peripheral IGF1 production defect.

We also attempted to use genetic methods to pinpoint a tissue source for the IGF1 deficiency. As was shown in the original submission, we showed that there was growth impairment and IGF1 deficiency in *Nestin-Cre; Arid1b^Fl/+^* mice. In contrast, there was no IGF1 or growth impairment in *Alb-Cre; Arid1b^Fl/+^* mice, supporting a central deficiency. Because the liver is thought to be the primary source of serum IGF1 production, it was perplexing to us why liver specific HET mice did not replicate the whole body *Arid1b^-/+^* mice. There are two main possibilities that we can speculate on. First, it is possible that a combination of central and peripheral defects in the GHRH-GH-IGF1 axis are required to fully recapitulate whole body *Arid1b* haploinsufficiency. Alternatively, the brain is the primary cause and source of the serum IGF1 deficiency. Future experiments will be required to precisely characterize the source of primary IGF1 deficiency in *Arid1b* HET mice. Nevertheless, these experiments provided some genetic evidence for a partial neural contribution to IGF1 deficiency, either in the form of reduced IGF1 production or in the form of inadequate GHRH-GH compensation for low IGF1 production.

As suggested by the reviewers, human patients can inform us about the biology of *ARID1B* haploinsufficiency. In a study with 60 *ARID1B* patients, height was shown to be significantly reduced (*[31]*). In another study, patients with *ARID1B* mutations demonstrated evidence for growth deficiency via partial GH deficiency (*[39]*). For this revision, we also obtained detailed clinical information from additional CSS patients, two with *ARID1B* mutations (from the www.arid1bgene.com database) and one with a mutation in *SMARCA4*, which encodes another SWI/SNF component. All three of these cases had deficiencies in the GH-IGF1 axis and clear beneficial responses to GH replacement therapy (Growth curves for the *ARID1B* patients are shown in Figure 5—figure supplement 1). It is unknown if these patients also had IGF1 and/or GHRH deficiency. Short stature is usually not further investigated after a diagnosis of CSS, thus it is likely that deficiencies in the GHRH-GH-IGF1 axis are under-diagnosed and rarely treated in this patient population.

In summary, extensive murine testing demonstrated a peripheral IGF1 deficiency with inadequate GHRH and GH compensation, contributing to growth impairment. In humans with *ARID1B* mutations, there is anecdotal evidence of GH deficiency and in most of those cases IGF1 and GHRH had not been examined. These findings provided the rationale for GH supplementation trials in our mice, where we found that HET mice were more responsive to GH supplementation.

*CSS is about 4x more common in females than males. Did the authors find any evidence to suggest the reason for this discrepancy that could relate to pituitary function, sex hormone alterations, or tolerance of mutation in males? Are there any difference in phenotypes between male and female Arid1b heterozygous and KO mice? Many tests were only done in male or female but not in both. In selection of genders for each test, are there any criteria that authors applied?*

Input from one of our clinical co-authors (Gijs Santen) makes us suspect that the historic reason that CSS was diagnosed more in females than in males rests partly upon the fact that some of the features (coarse face, hypertrichosis) are more likely to be considered abnormal in females. In the largest study to date about ARID1B patients there is no difference in the number of males/females: there are 30 males and 30 females in this study (*[31]*).

In our mouse model, we did not identify phenotypic differences between females and males in terms of growth and hormone measurements. Given the sheer numbers of mice needed to complete the study, we did not apply specific criteria for selection of mouse sex, we merely used what we could generate to run tests with larger cohorts. For USV vocalization we used both sexes and did not discern a difference between sexes. For grooming and marble burying, we only used females. For anxiety (elevated plus maze, dark light, open field) and juvenile social interaction testing, we used only males. We have not done extensive anxiety testing in females. Since it will take another several months for enough females to be at the correct age for anxiety testing, we can state explicitly in the text that our neuropsychiatric phenotypes were so far only validated for male mice.

*Reviewer #2:*

*Zhu and colleagues modeled ARID1B-opathies in mouse. Strong point of this paper are the multiple behavioral evidence presented by authors suggesting that Arid1b heterozygosity caused neuropsychiatric phenotypes. Overall, the paper is well-written and figures/data are presented in high quality. Although haploinsufficiency in the components of mammalian SWI/SNF complex is now well-known cause of neuropsychiatric disorders, the novelty lies in the reversal of growth defect by GH treatment.*

*Using the conditional KO animals, the authors speculated that the peripheral IGF1 deficiency due to low GH level is the underlying cause of short stature:*

*1) What is the primary pituitary defects of Arid1b heterozygous mouse?*

*2) How pituitary (defect) causes IGF1 deficiency in Arid1b heterozygote?*

Please see the response above.

References cited above.

1. J. L. Silverman, M. Yang, C. Lord, J. N. Crawley, Behavioural phenotyping assays for mouse models of autism. *Nat Rev Neurosci* 11, 490-502 (2010).

2. B. Greco *et al.*, Autism-related behavioral abnormalities in synapsin knockout mice. *Behav Brain Res* 251, 65-74 (2013).

3. J. Castro *et al.*, Functional recovery with recombinant human IGF1 treatment in a mouse model of Rett Syndrome. *Proc Natl Acad Sci U S A* 111, 9941-9946 (2014).

4. Y. X. Sun, P. Wang, H. Zheng, R. G. Smith, Ghrelin stimulation of growth hormone release and appetite is mediated through the growth hormone secretagogue receptor. *Proceedings of the National Academy of Sciences of the United States of America* 101, 4679-4684 (2004).

5. G. W. E. Santen, J. Clayton-Smith, A. B. C. S. S. c. the, The ARID1B phenotype: What we have learned so far. *American Journal of Medical Genetics Part C: Seminars in Medical Genetics* 166, 276-289 (2014).

6. Y. Yu *et al.*, De novo mutations in ARID1B associated with both syndromic and non-syndromic short stature. *BMC Genomics* 16, 701 (2015).